# Evaluation of SARS-CoV-2 in semen, seminal plasma, and spermatozoa pellet of COVID-19 patients in the acute stage of infection

**Lucie Delaroche**[1,2]*, **Mélanie Bertine**[3,4], **Pierre Oger**[1], **Diane Descamps**[3,4], **Florence Damond**[3,4], **Emmanuel Genauzeau**[1,2], **Philippe Meicler**[1], **Quentin Le Hingrat**[3,4], **Frédéric Lamazou**[1], **Rémi Gschwind**[3], **Etienne Ruppé**[3,5], **Benoit Visseaux**[3,4]

1 Ramsay Santé, Hôpital Privé de Parly 2, Institut Fertilité Maternité Parly 2, Le Chesnay, France, 2 Centre de Biologie Médicale, Hôpital Privé de Parly 2, Le Chesnay, France, 3 Université de Paris, INSERM, IAME, Paris, France, 4 AP-HP, Hôpital Bichat, Laboratoire de Virologie, Paris, France, 5 AP-HP, Hôpital Bichat, Laboratoire de Bactériologie, Paris, France

* luciedelaroche@yahoo.fr

**Data Availability Statement:** All relevant data can be found within the paper and Supporting Information files. The eCRF contains additional patients who were not included in the published

## Abstract

To date, there is limited information about the presence of SARS-CoV-2 in semen especially in the acute phase of the infection. While available data from cohort studies including a total of 342 patients in the acute or recovery phase of the infection are reassuring, one study mentioned detecting virus in the semen of 6/38 COVID-19 patients. Here we assessed SARS-CoV-2 presence in the semen of COVID-19 positive patients in the acute stage of infection, within 24 hours of the positive nasopharyngeal swabs. Semen, seminal plasma and spermatozoa pellet were screened for SARS-CoV-2 and manual or airborne contamination during semen sampling. Among the 32 COVID-19 volunteers, the median interval from the onset of symptoms to semen collection was 4 days [IQR: 0–8]. Only one presented positive SARS-CoV-2 PCR in semen and seminal plasma fractions, although the spermatozoa pellet was negative. Viral cultures were all negative. We observed slightly higher concentrations of bacterial DNA in the SARS-CoV-2 positive specimen than in all negative samples. The bacteria identified neither confirm nor rule out contamination by oropharyngeal secretions during collection. SARS-CoV-2 was rarely present in semen during the acute phase of the disease. This very rare situation could be connected to oral or manual contamination during semen collection. The possible presence of SARS-CoV-2 in semen calls for naso-pharyngeal viral testing and strict hygiene protocols during semen collection before assisted reproductive attempts.

## Introduction

In December 2019, a newly identified coronavirus named Severe Acute Respiratory Syndrome Coronavirus 2 (SARS-CoV-2) emerged in Wuhan, Hubei Province, China, resulting in the COVID-19 pandemic [1]. Although viral transmission occurs predominantly through

study. Thus, direct access by readers/others is not possible. Sequencing data were deposited on NCBI repository under the BioProject ID PRJNA721273.

**Funding:** Funding was obtained from Ramsay Santé and the French agency ANRS (National Agency for Research on AIDS and viral hepatitis). The funders had no role in study design, data collection and analysis, decision to publish, or preparation of the manuscript.

**Competing interests:** The authors have declared that no competing interests exist.

respiratory droplets, SARS-CoV-2 has been also isolated in blood samples, feces, and tears from patients with COVID-19, raising questions about viral shedding in other bodily fluids [2].

SARS-CoV-2 has a high affinity binding capability to the angiotensin-converting enzyme 2 (ACE2) in human cells, which is expressed in multiple organ systems, including the testes; and depends on transmembrane protease serine 2 (TMPRSS2) for cell entry and spread in host [3]. The ACE2 is predominantly enriched in spermatogonia, Leydig and Sertoli cells [4]. The co-expression of both ACE2 and TMPRSS2 genes was reported in spermatogonial stem cells, elongated spermatids, and in at least a small percentage of prostate hillock cells and in renal tubular cells [4–9]. The blood-testis barrier can be breached by viruses, especially in the presence of systemic and local inflammation [10–12]. Indeed, a wide range of viruses, such as Zika, Ebola, Influenza, Epstein Barr viruses that result in viremia can be detected in human semen [10, 13]. Moreover, specific male organs or cells could act as mid-term or long-term reservoirs for some of these viruses once infected. Taken all together, all these findings raise the question of the possible presence of SARS-CoV-2 in semen [14].

However, to date, there is limited data about the detection of SARS-CoV-2 in semen specimens. In a first study from China, Pan *et al.* investigated single semen samples from 34 recovering COVID-19 patients but did not find any trace of SARS-CoV-2 [15]. These results were confirmed by other cohort studies cumulating 342 COVID-19 patients [16–23]. It should be noted that 63 and 279 of these patients were in the acute (< 8 days from the onset of the symptoms or diagnosis) and recovery phases of the disease, respectively. Moreover, three studies reported no SARS-CoV-2 in the prostatic fluid of 89 COVID-19 patients [24]. In contrast, Li et al. observed 6 SARS-CoV-2 positive semen samples out of 38 semen samples from different COVID-19 patients, including 4 out of 15 hospitalized patients (26.7%) in the acute stage of infection (6 to 10 days after onset of symptoms), and 2 out of 23 patients (8.7%) recovering from the infection (12 to 16 days after onset of symptoms) [25]. However, the limits of detection and the threshold values were not described, and this observation has not been yet confirmed by any other report. Moreover, detailed virological assays and semen collection modalities were not provided. Possible contamination with RNA fragments from hands or respiratory droplets was not assessed.

Due to the few reports about the presence of SARS-CoV-2 data in semen to date and questions surrounding the infection risks of medically assisted reproduction while the COVID-19 pandemic is still ongoing, we aimed to determine whether SARS-CoV-2 could be detected in semen, seminal plasma, and spermatozoa pellet samples of COVID-19 positive patients in the acute phase of infection (≤ 8 days after the onset of symptoms).

## Results

A total of 132 male COVID-19 positive patients were asked to participate. Fifty-seven did not answer, 35 refused and 3 could not participate because of severe pathologies (Fig 1).

A total of 37 volunteers diagnosed positive with COVID-19 were included from August 2020 to April 2021. One patient failed to provide a semen sample at hospital. Four patients in the recovery phase (interval from onset of symptoms to semen collection between 13 and 45 days) were excluded from the analysis. Accordingly, 32 COVID-19 patients in the acute phase of infection were included and analyzed in this study.

### Patient characteristics

The 32 participants had a mean age of 38.8 ± 10.9 years with mean body mass index (BMI) 26.6 ± 4.3 kg/m$^2$ (S1 Table). Six patients (9%) were current smokers and three others (9%)

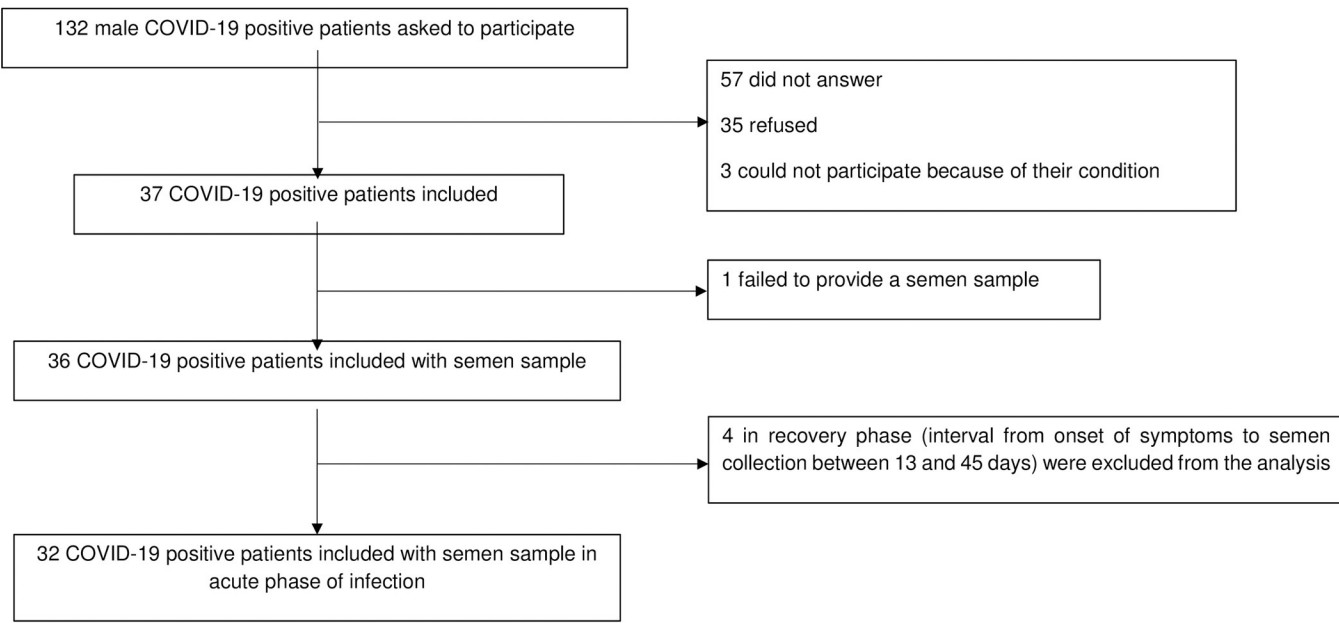

**Fig 1. Flow chart of the study.** From a total of 132 male COVID-19 positive patients requested to participate, 32 patients in the acute phase of infection who provided a semen sample were included.

former smokers. Two patients (6%) suffered from arterial hypertension, two (6%) from chronic respiratory disease, one (3%) from vision disorders, one (3%) from type 2 diabetes and one (3%) from anxiety. One patient (3%) was infected with the human immunodeficiency virus-1 (HIV-1) and was being treated with antiretrovirals. One patient (3%) had undergone vasectomy. Twenty-seven (84%) patients were symptomatic, with moderate symptoms. None mentioned orchitis-related symptoms. No specific COVID-19 treatment was administered, only analgesics for 12 patients (38%). One patient (3%) received antibiotics. The other 5 patients (16%) were asymptomatic (contact cases).

The median interval from onset of symptoms to providing semen collection was 4 days [IQR: 0–8] and the median time between the nasopharyngeal swab sample and the collection of a semen sample was 1 day [0–1]. For three patients (n˚1, 2 and 22), semen was collected at home.

## SARS-CoV-2 in nasopharyngeal samples

The cycle threshold (Ct) in nasopharyngeal swabs ranged from 14.5 to 35.9. The screening for recent variants of concerns revealed a 501Y.V1 variant (B.1.1.7, commonly designated as the "UK variant") in 10 patients (Table 1).

## SARS-CoV-2 in semen samples

The semen samples (semen, seminal plasma, and spermatozoa pellet) of each COVID-19 patient were screened for SARS-CoV-2 by RT-PCR, except for the vasectomized patient, for whom the only semen sample, which corresponded to seminal plasma, was analyzed.

The semen samples of patients n˚14 and n˚25, the seminal plasma of patient n˚25 and the spermatozoa pellet of patient n˚18 were uninterpretable because of the presence of inhibitors. No SARS-CoV-2 was detected in any samples, except for those of one patient (Table 1).

**Table 1. Test results for SARS-CoV-2 in nasopharyngeal and semen samples of the 32 enrolled patients.**

| Patient | Time between onset of symptoms and nasopharyngeal swab (days) | Time between nasopharyngeal samples and semen (days) | SARS-CoV-2 detection in the nasopharyngeal swab | RT-PCR system | Cycle thresholds (Ct) in the nasopharyngeal swab | SARS-CoV-2 detection | | |
|---|---|---|---|---|---|---|---|---|
| | | | | | | Semen | Seminal plasma | Spermatozoa pellet |
| 1 | 5 | 1 | Positive | ELITe InGenius | RdRp gene = 18.2; N gene = 19.5; E gene = 16.8 | Negative | Negative | Negative |
| 2 | 5 | 1 | Positive | ELITe InGenius | RdRp gene = 23.0; N gene = 24.6; E gene = 22.2 | **Positive** | **Positive** | Negative |
| 3 | 7 | 0 | Positive | Altona | E gene = 33.6 | Negative | Negative | Negative |
| 4 | 7 | 0 | Positive | Altona | E gene = 18.0 | Negative | Negative | Negative |
| 5 | 3 | 1 | Positive | Altona | E gene = 18.4 | Negative | Negative | Negative |
| 6 | 2 | 0 | Positive | Altona | E gene = 18.4 | Negative | Negative | Negative |
| 7 | 6 | 1 | Positive | Altona | E gene = 26.9 | Negative | Negative | Negative |
| 8 | 5 | 0 | Positive | Altona | E gene = 16.0 | Negative | Negative | Negative |
| 9 | 3 | 1 | Positive | Altona | E gene = 16.3 | Negative | Negative | Negative |
| 10 | 4 | 0 | Positive | Altona | E gene = 16.0 | Negative | Negative | Negative |
| 11 | 3 | 1 | Positive | Altona | E gene = 18.0 | ND[b] | Negative | ND[b] |
| 12 | 5 | 1 | Positive | ELITe InGenius | RdRp gene = 18.9; N gene = 19.0; E gene = 19.2 | Negative | Negative | Negative |
| 13 | 5 | 1 | Positive | ELITe InGenius | RdRp gene = 14.4; N gene = 19.0; E gene = 16.6 | Negative | Negative | Negative |
| 14 | 5 | 1 | Positive | Altona | E gene = 22.7 | Uninterpretable (Inhibitors) | Negative | Negative |
| 15 | 1 | 1 | Positive | Altona | E gene = 20.4 | Negative | Negative | Negative |
| 16 | 0 | 0 | Positive | Altona | E gene = 26.1 | Negative | Negative | Negative |
| 17 | 3 | 0 | Positive | Altona | E gene = 24.6 | Negative | Negative | Negative |
| 18 | 6 | 0 | Positive | Altona | E gene = 25.0 | Negative | Negative | Uninterpretable (Inhibitors) |
| 19 | 2 | 0 | Positive[a] | Altona | E gene = 27.5 | Negative | Negative | Negative |
| 20 | 2 | 1 | Positive | Altona | E gene = 33.3 | Negative | Negative | Negative |
| 21 | 4 | 0 | Positive[a] | Altona | E gene = 17.8 | Negative | Negative | Negative |
| 22 | 2 | 0 | Positive[a] | Altona | E gene = 17.6 | Negative | Negative | Negative |
| 23 | 8 | 0 | Positive[a] | Altona | E gene = 27.5 | Negative | Negative | Negative |
| 24 | 1 | 0 | Positive[a] | CFX96 | N1 gene = 25; N2 gene = 25 | Negative | Negative | Negative |
| 25 | 3 | 1 | Positive[a] | Altona | E gene = 30.4 | Uninterpretable (Inhibitors) | Uninterpretable (Inhibitors) | Negative |
| 26 | 1 | 1 | Positive | Altona | E gene = 35.9 | Negative | Negative | Negative |
| 27 | 6 | 1 | Positive[a] | Altona | E gene = 32.7 | Negative | Negative | ND[b] |
| 28 | 1 | 0 | Positive[a] | ABI PRISM 7500 | N gene = 15.0 | Negative | Negative | Negative |
| 29 | 4 | 1 | Positive[a] | Altona | E gene = 20.6 | Negative | Negative | Negative |
| 30 | 4 | 0 | Positive[a] | Altona | E gene = 21.3 | Negative | Negative | Negative |
| 31 | 1 | 0 | Positive | CFX96 | N1 gene = 18; N2 gene = 19 | Negative | Negative | Negative |

*(Continued)*

**Table 1.** (Continued)

| Patient | Time between onset of symptoms and nasopharyngeal swab (days) | Time between nasopharyngeal samples and semen (days) | SARS-CoV-2 detection in the nasopharyngeal swab | RT-PCR system | Cycle thresholds (Ct) in the nasopharyngeal swab | SARS-CoV-2 detection | | |
|---|---|---|---|---|---|---|---|---|
| | | | | | | Semen | Seminal plasma | Spermatozoa pellet |
| 32 | 7 | 0 | Positive | CFX96 | N1 gene = 31; N2 gene = 32 | Negative | Negative | Negative |

[a] Presence of mutations suggestive of the B1.1.7 (N501Y.V1—UK variant)

[b] ND: Not done

The time between onset of symptoms and semen collection (days) and the time between semen and nasopharyngeal samples (days) are presented for each of the 32 enrolled COVID-19 patients. The cycle threshold (Ct) in the nasopharyngeal swabs and the SARS-CoV-2 detection in semen samples are described.

Indeed, patient n˚2 had positive SARS-CoV-2 detection in both semen sample and seminal plasma fraction. The Ct from the RealStar® assay were at 27.6 and 29.9 with gene E and 27.6 and 29.7 with gene S for the semen and the seminal plasma fraction, respectively. When confirmed the following day on the Simplexa® assay, Ct were at 31.6 and 30.2 with gene S and 31.7 and 30.1 with gene ORF1ab for the semen and the seminal plasma fraction, respectively. The spermatozoa pellet was negative for SARS-CoV-2 with both PCR assays.

## Bacterial analysis of semen samples

As the patient with positive SARS-CoV-2 semen and seminal plasma samples performed his sperm collection at home, we assessed possible manual or droplet contamination during the semen sampling by analyzing bacterial DNA presence. Bacterial DNA was detected in the positive SARS-CoV-2 semen at higher concentrations ($6.7 \times 10^3$ 16S copies/µL of DNA) than in the 31 negative SARS-CoV-2 semen samples (mean of $1.2 \times 10^2$ 16S copies/µL of DNA (Fig 2). Following these results, the bacterial composition of the SARS-CoV-2 positive semen and seminal plasma samples was analyzed using amplicon sequencing on Flongle flow cells. The main bacterial genera were *Haemophilus and Finegoldia*. *Fusobacterium*, *Actinobacillus*, *Prevotella*, *Peptoniphilus* and *Streptococcus* were also evidenced. No *Rothia* was detected.

## Viral culture

To assess whether the detected viruses were infectious, the positive SARS-CoV-2 sample was cultivated on Vero E6 cells without any sign of viral replication. It is important to note that the sample was frozen twice before viral culture testing.

## Discussion

This study reported no evidence of SARS-CoV-2 in the semen, seminal plasma, and spermatozoa pellet samples of 31/32 (97%) enrolled COVID-19 patients in the acute phase of the disease, in line with previous studies [22–24]. However, we detected the presence of SARS-CoV-2 within the semen sample and the seminal plasma fraction of one COVID-19 patient (3%) in the acute stage of the disease. Although the patient confirmed having followed the aseptic semen collection method, bacterial DNA was detected at higher concentrations in the positive SARS-CoV-2 semen sample than in the negative SARS-CoV-2 semen samples of all other included patients. The main bacterial genera observed in the positive SARS-CoV-2 semen sample were *Haemophilus*, *Finegoldia*, and *Fusobacterium*. These bacteria are commonly found in the oropharyngeal microbiota, but they have also been described in the sperm [26]. Furthermore, we also found in the SARS-CoV-2 positive semen specimen *Prevotella* and

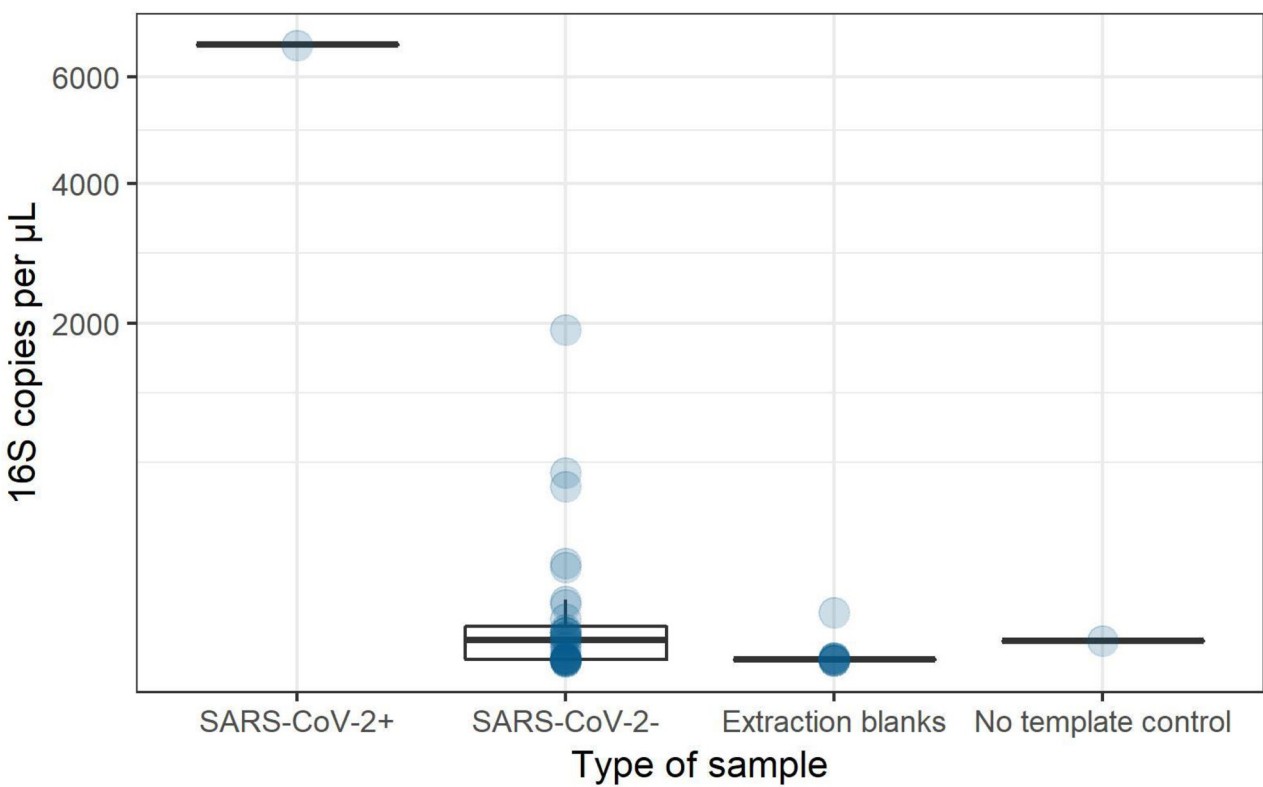

**Fig 2. 16S rRNA gene copy number per µL of DNA in SARS-CoV-2 positive or negative native semen samples.** Extraction blanks are DNA extraction no template controls. No template control was done by adding water instead of DNA in the qPCR reaction mix.

*Streptococcus* but no *Rothia*, which are usually found as the dominant genus in the oropharynx of human subjects [27]. Hence, the bacteria found in the SARS-CoV-2 positive semen specimen can neither confirm nor strictly rule out a weak contamination of the semen sample by oropharyngeal secretions. Nonetheless, the bacteria we found do not support contamination by the skin microbiota [27]. The spermatozoa pellet was SARS-CoV-2 negative, which is reassuring regarding viral safety procedures, despite the Ct values of the two positive samples demonstrating non-negligible viral loads. Moreover, the viral culture was negative, which suggest the absence of a strong infection risk at these levels of viral load. However, as other cell models have recently been described as more sensitive than Vero E6, due to the low number of positive samples and as the sample was not tested fresh but after freeze-thaw steps, we should remain cautious when interpreting this observation [28].

Analogous to our SARS-CoV-2 positive semen specimen, Li et al. detected SARS-CoV-2 among 4/15 and 2/23 patients in the acute phase and after the clinical recovery from COVID-19, respectively [25]. All these patients were tested in a short time frame after the onset of symptoms (from 6 to 16 days for semen positive patients). However, the work of Li et al. does not provide detailed virological results of their findings, notably the virological assays and viral load estimation, and it did not test for oropharyngeal contamination.

The question of a possible passage of SARS-CoV-2 in the sperm is still debated. The expression of ACE-2, TMPRSS2, and CD147 receptors in the testes, epididymis, prostrate and seminal vesicles has been reported [29], which support the hypothesis of a potential entry of the SARS-CoV-2. Moreover, in the testes, epididymis and seminal vesicles, the expression of lysosomal cathepsins (CTSB/CTSL) and/ neuropilin-1 (NRP-1) which also promote viral invasion

have been evidenced. Altogether, these findings led to speculation that gonads offered the proper ground for SARS-CoV-2 replication. Besides, the report of SARS-CoV-2-induced orchitis suggested that testicular infection might damage the testis-blood barrier and permit viral shedding into semen. However, the lack of expression of the TMPRSS2 modulatory protein in testicular cells and sperm argues against the hypothesis that gametes transmit SARS-CoV-2 given that TMPRSS2 is required for SARS-CoV-2 cell entry [24]. Hence, large-scale experiments are still needed to determine the risks of transmission of SARS-CoV-2 to semen.

This work suggests that the use of discontinuous density gradient centrifugation could eliminate the presence of SARS-CoV-2 from positive semen samples since the spermatozoa pellet was negative, as has been described for other sexually transmitted viruses such as the Human Immunodeficiency Virus (HIV) [30]. Unfortunately, the patient with the SARS-CoV-2 positive semen did not come back to provide new nasopharyngeal and semen samples one month after the first ones as scheduled, which would have enabled us to assess for the disappearance of SARS-CoV-2 in the sperm.

The impact of SARS-CoV-2 on male reproductive function, including fertility and testicular endocrine functions, as well as its infectiousness, still remains to be determined [31, 32]. Recent studies showed that COVID-19 could impair male fertility by inducing orchitis, and decreasing testosterone levels, sperm counts and motility [11, 12, 16, 17, 33–37]. However, a recent review of epidemiological investigations, molecular receptor identification and detection studies of SARS-CoV-2 RNA in testicular biopsies, semen and prostatic fluids, vaginal fluids and cervical smears suggest that COVID-19 is not a sexually transmitted disease [24].

Based on these preliminary results and consistent with prior findings, the possible presence of SARS-CoV-2 in semen cannot be excluded, especially during the acute phase of the disease. However, this situation seems very rare and may be associated with oral or manual contamination during semen collection. We should remain cautious in assisted reproductive technology centers: viral testing using nasopharyngeal swabs before IVF attempts and strict disinfection protocols at semen collection should be recommended for increasing viral safety.

## Materials and methods

COVISPERM ("COVID detection in SPERM") is a prospective observational study assessing the presence of SARS-CoV-2 in semen samples of positive COVID-19 patients. This study (NCT04460534) was approved by the French Ethics committee Sud Mediterranean III on the 5th of May 2020 (ANSM 2020-A01206-33). The research was carried out in the Clinical Biology Laboratory of the Ramsay Santé Private Hospital of Parly 2, Le Chesnay, France. Written informed consent was obtained from patients before inclusion, and all experiments were performed in accordance with relevant named guidelines and regulations.

Male outpatients, aged between 18 and 65 years old and diagnosed positive for SARS-CoV-2 from nasopharyngeal samples whether they were asymptomatic or had moderate symptoms were invited to participate. By moderate symptoms, we meant ambulatory or hospitalized patients in acute phase of the infection ($\leq$ 8 days after the onset of symptoms) without signs of respiratory severity able to come to the laboratory. For asymptomatic patients, the date of the first COVID-19 positive nasopharyngeal test was considered as the date of onset of symptoms. Enrolled patients were asked to provide a semen sample within 24 hours after their positive SARS-CoV-2 nasopharyngeal sample screening. The patients' medical characteristics including demographic data, potential comorbidities, chronic pathologies, usual treatments, and symptoms were recorded.

Semen collection was obtained by masturbation in a laboratory room or at home according to the patient's clinical condition and possible difficulties in collecting semen. In the latter

case, the material for the collection (sterile container, disinfectant wipes, sterile water pods) was provided by the laboratory, and the patients were asked to bring their sample at room temperature to the laboratory within the next hour. Before semen collection, hygiene procedures were explained to the patients (hand washing with soap, penis washing with disinfectant detergent, rinsing with sterile water) to avoid virus contamination from other non-semen sources. Also, patients had to wear a mask during semen collection.

Freshly collected semen was liquefied at room temperature for between 30 minutes and 1 hour. Before preparing the sperm, one aliquot of 200 μl semen sample was prepared for viral testing. Then, the remaining semen sample was centrifuged at 350g for 20 min on a discontinuous ISolate® (IrvineScientific, USA) density gradient using a 40% (v/v) density top layer and a 80% (v/v) density lower layer [38]. The supernatant (i.e. the seminal plasma) was then removed from the sperm pellet and aliquoted for viral testing. The sperm clot was resuspended in 5 mL MHM® (IrvineScientific, USA) and centrifuged at 200g for 7 min. The final pellet was resuspended in 0.6 mL MHM® and aliquoted for viral testing. All three aliquots (semen, seminal plasma, and spermatozoa pellet) from each patient were stored in double packaging at -20˚C before viral analyses.

Detection of SARS-CoV-2 in nasopharyngeal swab samples was performed by real-time reverse transcriptase-polymerase chain reaction (RT-PCR) using either the SARS-CoV-2 ELITe MGB® Kit on an ELITe InGenius® system (ELITech Group) detecting the RdRp, the N and the E genes, or the RealStar® SARS-CoV-2 RT-PCR kit (Altona) detecting the E gene, after RNA extraction using the MagNA Pure LC 2.0 System (Roche) with the Total Nucleic Acid Isolation kit—Large Volume (Roche), providing a limit of detection of 625 copies/mL [39], depending on the availability of the instruments. The SARS-CoV-2 Droplet Digital PCR ® Kit detecting the N1 and the N2 genes on a CFX96 (Bio rad) and the SARS-CoV-2 GSD NovaPrime® Kit detecting two specific regions of the N gene on an ABI PRISM 7500 (ThermoFisher) were also used for the detection of SARS-CoV-2 in nasopharyngeal swabs.

Patients were screened for the presence of mutations suggestive of the B1.1.7 (N501Y.V1—UK), B1.351 (N501Y.V2—South African) and B.1.1.28 (N501Y.V3 –Brazilian) variants. This screening was performed using specific PCR detection of N501Y and E484K mutations using corresponding VirSNiP Assays (Tib MolBiol) and the multi-target RT-qPCR TaqPath® COVID-19 diagnostic test (ThermoFisher), allowing the 69–70 del from the B.1.1.7 variant to be evidenced.

Semen samples were transported at -20˚C and analyzed in the Virology Laboratory of Bichat Hospital, Paris, France. Detection of SARS-CoV-2 in semen samples was performed by RT-qPCR using the RealStar® SARS-CoV-2 RT-PCR kit (Altona) as described for the nasopharyngeal samples. Positive samples were confirmed using the Simplexa® COVID-19 Direct kit (DiaSorin Molecular) providing a limit of detection of 316 copies/mL [40]. All assays were performed according to the manufacturer recommendations.

To assess possible manual and oropharyngeal contamination of the semen samples, bacterial DNA presence was analyzed by qPCR targeting the V8-V9 region of the 16S rRNA encoding gene. Forward (5' CGGTGAATACGTTCCCGG 3') and reverse (5' TACGGCTACCTTGTTACGACTT 3') primers were mixed with the KAPA SYBR® FAST qPCR Master Mix (2X) together with 4 μL of total nucleic acid extract. The reaction was carried out on a LightCycler® 480 II (Roche) using the following program: 95˚C 3', (95˚C 10 min, 55˚C 20 sec, 72˚C 30 sec) x40. Results were analyzed using LightCycler® 480 SW 1.5.1 software. Then, the bacterial composition of the samples showing bacterial DNA presence was determined. First, a PCR that better targeted the V3-V4 region (yielding a longer fragment of the 16S rRNA gene and thus spanning more hypervariable regions) was made using KAPA HiFi HotStart Ready Mix kit. When visible after gel electrophoresis, amplicons were sequenced

on a MinION device (Oxford Nanopore Technologies). A library was made using the Rapid Barcoding Kit (RBK004) and 120 ng of DNA from each sample (semen and seminal plasma fraction) were loaded on a Flongle flow cell (FLO-FLG001). Reads were assigned a taxonomy using the EPI2ME software [41].

Viral isolation was performed in a BSL-3 laboratory. Briefly, 100 μL of tested sample was diluted into 900 μL of Dulbecco's Modified Eagle Medium (DMEM, Gibco), and filtered through a 0.45 μm filter (Sartorius). Then 500 μL of the filtered material was inoculated to 50,000 Vero E6 cells (reference CRL-1586, ATCC) plated in a 24-well plate. After one hour at 37˚C, 500 μL of DMEM containing 4% of FBS (Fetal Bovine Serum, Gibco) was added to each well. At day 6 post-infection, wells were screened for cytopathogenic effect and by RT-PCR (RealStarⓇ SARS-CoV-2 RT-PCR, Altona).

## Supporting information

**S1 Table. Clinical characteristics of the 32 enrolled COVID-19 patients.**
(DOCX)

**S1 Data.**
(XLSX)

## Acknowledgments

The authors thank Dr. CHIARELLI, Mrs. COURTIN, Mr. LOCRET, Mrs. DELAGE and Mr. SAHBANE for their invaluable help and advice throughout the study; and Mrs. PETITJEAN and Mrs. VENTURINI for assistance with bioinformatics and statistics; and ACOLAD for English revision.

## Author Contributions

**Conceptualization:** Lucie Delaroche.

**Investigation:** Lucie Delaroche, Mélanie Bertine, Pierre Oger, Etienne Ruppé, Benoit Visseaux.

**Supervision:** Lucie Delaroche.

**Validation:** Lucie Delaroche, Mélanie Bertine, Pierre Oger, Diane Descamps, Florence Damond, Emmanuel Genauzeau, Philippe Meicler, Quentin Le Hingrat, Frédéric Lamazou, Rémi Gschwind, Etienne Ruppé, Benoit Visseaux.

**Writing – original draft:** Lucie Delaroche.

**Writing – review & editing:** Lucie Delaroche, Pierre Oger, Etienne Ruppé, Benoit Visseaux.

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
