## [Decision Letter · Decision Letter 0]

13 Sep 2021

PONE-D-21-24028Evaluation of SARS-CoV-2 in semen, seminal plasma, and final fraction with spermatozoa of COVID-19 patients at acute stage of infectionPLOS ONE

Dear Dr. DELAROCHE,

Thank you for submitting your manuscript to PLOS ONE. After careful consideration, we feel that it has merit but does not fully meet PLOS ONE’s publication criteria as it currently stands. Therefore, we invite you to submit a revised version of the manuscript that addresses the points raised during the review process.

All reviewers find the work of interest even if it is a description paper. There is some information that deserved to be provided before publication as indicated by the reviewers. Please try to answer to all the comments and include the points in the article. As noted the introduction needs some editing to shortened it. Try also to provide a more synthetic table to describe the final test if possible in a general effort for an easy-to-read paper.

We look forward to receiving your revised manuscript.

Kind regards,

Pierre Roques, Ph.D.

Academic Editor

PLOS ONE

Journal Requirements:

Reviewers' comments:

Reviewer's Responses to Questions

**Comments to the Author**

1. Is the manuscript technically sound, and do the data support the conclusions?

Reviewer #1: Yes

Reviewer #2: Yes

Reviewer #3: Yes

2. Has the statistical analysis been performed appropriately and rigorously? 

Reviewer #1: Yes

Reviewer #2: N/A

Reviewer #3: Yes

3. Have the authors made all data underlying the findings in their manuscript fully available?

Reviewer #1: Yes

Reviewer #2: Yes

Reviewer #3: Yes

4. Is the manuscript presented in an intelligible fashion and written in standard English?

Reviewer #1: No

Reviewer #2: Yes

Reviewer #3: Yes

5. Review Comments to the Author

Reviewer #1: Reviewer’s report

Title: Evaluation of SARS-CoV-2 in semen, seminal plasma, and final fraction with spermatozoa of COVID-19 patients at acute stage of infection

Date: 31 August 2021

Manuscript ID: PONE-D-21-24028

Reviewer’s report:

In the manuscript titled “: Evaluation of SARS-CoV-2 in semen, seminal plasma, and final fraction with spermatozoa of COVID-19 patients at acute stage of infection” the authors evaluated the presence of SARS-CoV-2 in semen, seminal plasma and pallet samples of COVID-19 positive patients in the acute phase of infection. The topic could be of interest however several concerns need to be raised:

1- I suggest changing the “native semen” to semen which is more common.

2- Do the authors mean pallet after spinning the semen sample by “final fraction containing spermatozoa”? If yes change it to pallet.

3- Please write down these expressions: ESHRE, IFFS, ASRM in the instruction.

4- Introduction is too long. I recommend shortening it.

5- Please clarify and explain which criteria did the authors considered as an acute phase and moderate symptom?

6- Discussion lacks depth (it is purely descriptive).

7- It is noted that this manuscript needs a professional technical English editing service paying particular attention to English grammar and sentence structure.

Reviewer #2: The article from L. Delaroche et al. addresses the question of the presence of SARS-Cov-2 in semen from COVID-19 patients in the acute stage of infection. This study is an observational study.

Thirty-two men were included in the study and the virus was detected by RT-qPCR using the RealStar RT-PCR kit in native semen, seminal plasma and 80% fraction obtained after centrifugation of the semen on discontinuous density gradients. Only one volunteer presented positive SARS-Cov-2 RT-PCR in semen and seminal plasma, 80%fraction were negative. Following bacterial investigation of semen samples, the authors suggested that the SARS-Cov-2 presence in semen from one volunteer is more a consequence of oral contamination than e semen SARS-Cov-2 infection.

I have several comments:

I did not understand the reasons of semen home collection in 3 patients. The authors explain this: “according to the patient’s clinical condition”. This point needs to be clarified as 2 patients were symptomatic and the last was asymptomatic.

The semen investigation methods are not totally comprehensives. Linea 261: is the supernatant obtained from native semen centrifugation or from density gradient centrifugation? If is the first case, I suggest to present before “the remaining semen samples…(linea 258).

Table 1 and 2 reported the result characteristics in detail. For the reader, I think that table 1 could be more synthetic.

One patient had a positive semen. The patient performed semen probe at home and the precise bacterial study suggest a possible contamination from oropharyngeal secretions. In discussion, the authors mentioned the “The bacteria found can neither confirm nor strictly rule out a weak contamination of the semen sample by oropharyngeal secretions”. Indeed, higher concentrations of bacterial DNA were found in this specimen than in others but we did not know the levels which allows to conclude oral contamination in one and no in others. The sentence in abstract “the bacteria identified do not clearly rule out contamination by oropharyngeal secretions” could be modified according to “neither confirm nor rule out contamination” according to discussion.

Reviewer #3: A study with a final sample size of 32 COVID-19 positive volunteers aimed to determine whether SARS-CoV-2 could be detected in native semen, seminal plasma, and final fraction with spermatozoa samples. SARS-CoV-2 was detected in only one volunteer’s native semen and seminal plasma samples. The results are primarily descriptive.

Minor revisions:

1- Line 104: Indicate the type of summary statistics provided for age.

2- Patient characteristics: In addition to the frequencies, state the corresponding percentages.

3- Line 186: Provide the corresponding percentage and a 95% confidence interval for the 31/32 result.

6. PLOS authors have the option to publish the peer review history of their article (what does this mean?). If published, this will include your full peer review and any attached files.

Reviewer #1: **Yes: **kajal Khodamoradi

Reviewer #2: **Yes: **Bujan Louis

Reviewer #3: No

---

## [Author Response · Author response to Decision Letter 0]

9 Oct 2021

Dear Reviewers, 

we sent the manuscript with revisions. 

Best regards

Lucie Delaroche

---

## [Decision Letter · Decision Letter 1]

4 Nov 2021

Evaluation of SARS-CoV-2 in semen, seminal plasma, and spermatozoa pellet of COVID-19 patients in the acute stage of infection

PONE-D-21-24028R1

Dear Dr. DELAROCHE,

We’re pleased to inform you that your manuscript has been judged scientifically suitable for publication and will be formally accepted for publication once it meets all outstanding technical requirements.

Kind regards,

Pierre Roques, Ph.D.

Academic Editor

PLOS ONE

Additional Editor Comments (optional):

Reviewers' comments:

Reviewer's Responses to Questions

**Comments to the Author**

1. If the authors have adequately addressed your comments raised in a previous round of review and you feel that this manuscript is now acceptable for publication, you may indicate that here to bypass the “Comments to the Author” section, enter your conflict of interest statement in the “Confidential to Editor” section, and submit your "Accept" recommendation.

Reviewer #1: All comments have been addressed

Reviewer #3: All comments have been addressed

2. Is the manuscript technically sound, and do the data support the conclusions?

Reviewer #1: Yes

Reviewer #3: (No Response)

3. Has the statistical analysis been performed appropriately and rigorously? 

Reviewer #1: Yes

Reviewer #3: (No Response)

4. Have the authors made all data underlying the findings in their manuscript fully available?

Reviewer #1: Yes

Reviewer #3: (No Response)

5. Is the manuscript presented in an intelligible fashion and written in standard English?

Reviewer #1: Yes

Reviewer #3: (No Response)

6. Review Comments to the Author

Reviewer #1: (No Response)

Reviewer #3: (No Response)

7. PLOS authors have the option to publish the peer review history of their article (what does this mean?). If published, this will include your full peer review and any attached files.

Reviewer #1: **Yes: **kajal khodamoradi

Reviewer #3: No

---

## [Editor Report · Acceptance letter]

3 Dec 2021

PONE-D-21-24028R1 

Evaluation of SARS-CoV-2 in semen, seminal plasma, and spermatozoa pellet of COVID-19 patients in the acute stage of infection 

Dear Dr. Delaroche:

I'm pleased to inform you that your manuscript has been deemed suitable for publication in PLOS ONE. Congratulations! Your manuscript is now with our production department. 

Kind regards, 

on behalf of

Dr. Pierre Roques 

Academic Editor

PLOS ONE